# "Putting the power back into community": A mixed methods evaluation of a chronic hepatitis B training course for the Aboriginal health workforce of Australia's Northern Territory

Kelly Hosking[1,2,3]*, Teresa De Santis[3], Emily Vintour-Cesar[1,2], Phillip Merrdi Wilson[3], Linda Bunn[3], George Garambaka Gurruwiwi[2], Shiraline Wurrawilya[3], Sarah Mariyalawuy Bukulatjpi[4], Sandra Nelson[3], Cheryl Ross[2], Kelly-Anne Stuart-Carter[5], Terese Ngurruwuthun[4], Amanda Dhagapan[4], Paula Binks[2], Richard Sullivan[2,6], Linda Ward[2], Phoebe Schroder[7], Jaclyn Tate-Baker[8], Joshua S. Davis[2,9], Christine Connors[1,3], Jane Davies[2,8], On behalf of the Hep B PAST partnership[¶]

1 Public Health Directorate, Office of the Chief Health Officer, Northern Territory Health, Northern Territory, Australia, 2 Global and Tropical Health Division, Menzies School of Health Research, Charles Darwin University, Darwin, Northern Territory, Australia, 3 Population and Primary Health Care Branch, Top End Health Service, Northern Territory Health, Northern Territory, Australia, 4 Miwatj Aboriginal Health Corporation, Nhulunbuy, East Arnhem Land, Northern Territory, Australia, 5 Centre for Disease Control, Northern Territory Health, Alice Springs, Northern Territory, Australia, 6 UNSW School of Clinical Medicine, St George & Sutherland Campus, Jannali, NSW, Australia, 7 Australasian Society for HIV, Viral Hepatitis and Sexual Health Medicine, Sydney, NSW, Australia, 8 Department of Infectious Diseases, Royal Darwin and Palmerston Hospital, Northern Territory Health, Darwin, Northern Territory, Australia, 9 School of Medicine and Public Health, University of Newcastle, Callaghan, NSW, Australia

☯ These authors contributed equally to this work.
¶ The complete membership of the Hep B PAST partnership can be found in the acknowledgments
* kelly.hosking@menzies.edu.au

**Data Availability Statement:** Ethical and privacy considerations restrict public access to the data

## Abstract

### Background

Chronic hepatitis B (CHB) is endemic in the Aboriginal and Torres Strait Islander population of Australia's Northern Territory. Progression to liver disease can be prevented if holistic care is provided. Low health literacy amongst health professionals is a known barrier to caring for people living with CHB. We co-designed and delivered a culturally safe "Managing hepatitis B" training course for the Aboriginal health workforce. Here, we present an evaluation of the course.

### Objectives

1. To improve course participants CHB-related knowledge, attitudes, and clinical practice.
2. To evaluate the "Managing hepatitis B" training course.
3. To enable participants to have the skills and confidence to be part of the care team.

collected and analysed in this study. Data may be available for reasonable requests through to the Hep B PAST steering committee, email: Hepbpast@menzies.edu.au. Course materials and guidelines of use can be requested through the corresponding author.

**Funding:** The Australian National Health and Medical Research Council (NHMRC) supported this work. This research is part of the Hep B PAST project, which receives an NHMRC partnership grant, GNT1151837. KH is undertaking a PhD and has an NHRMC scholarship, GNT1190918. The funders had no role in study design, data collection and analysis, decision to publish, or preparation of the manuscript. There was no additional external funding received for this study.

**Competing interests:** The authors have declared that no competing interests exist.

**Abbreviations:** AHP, Aboriginal Health Practitioner; ACW, Aboriginal Community Worker; CHB, Chronic hepatitis B; COVID-19, SARS-CoV-2; HBV, Hepatitis B virus; HCC, Hepatocellular carcinoma; NHMRC, National Health and Medical Research Centre; NT, Northern Territory; PAR, Participatory action research; PHC, Primary health care.

## Methods

We used participatory action research and culturally safe principles. We used purpose-built quantitative and qualitative evaluation tools to evaluate our "Managing hepatitis B" training course. We integrated the two forms of data, deductively analysing codes, grouped into categories, and assessed pedagogical outcomes against Kirkpatrick's training evaluation framework.

## Results

Eight courses were delivered between 2019 and 2023, with 130 participants from 32 communities. Pre- and post-course questionnaires demonstrated statistically significant improvements in all domains, $p < 0.001$ on 93 matched pairs. Thematic network analysis demonstrated high levels of course acceptability and significant knowledge acquisition. Other themes identified include cultural safety, shame, previous misinformation, and misconceptions about transmission. Observations demonstrate improvements in post-course engagement, a deep understanding of CHB as well as increased participation in clinical care teams.

## Conclusions

The "Managing hepatitis B" training course led to a sustained improvement in the knowledge and attitudes of the Aboriginal health workforce, resulting in improved care and treatment uptake for people living with CHB. Important non-clinical outcomes included strengthening teaching and leadership skills, and empowerment.

## Introduction

### Epidemiology and issue

Chronic hepatitis B (CHB) infection is a major public health issue, with an estimated global prevalence of 3.9% [1]. CHB disproportionately affects First Nations populations worldwide [2]. The Northern Territory (NT) has the highest prevalence of CHB in Australia (1.84% in comparison to 0.86% nationally) [3]. Australia's First Nations people, respectfully referred to as Aboriginal and Torres Strait Islander people, experience a CHB prevalence 2 to 6 times higher than non-Indigenous people [4,5], representing approximately 70% of those affected in the NT [6]. It is estimated that of those living with CHB, 25% (15–40%) will die from liver disease, namely hepatocellular carcinoma (HCC) - primary liver cancer, and cirrhosis [7,8] without appropriate care. Recent gains in closing the gap in life expectancy between Aboriginal and Torres Strait Islander and non-Indigenous peoples in the NT were primarily attributed to a decrease in cancer deaths, with the exception of liver cancer, which remains high [9]. The adverse outcomes of CHB can be prevented with engagement in care, monitoring, and publicly funded treatment [8,10,11].

### Elimination of CHB as a public health threat and the cascade of care

Highly effective prevention and treatment interventions, such as vaccination, antenatal care and antiviral medication, have made the elimination of CHB a realistic possibility [12].

International sustainable development goals and the World Health Organization (WHO) aim to eliminate hepatitis B virus (HBV) as a global public health threat by 2030, highlighting the urgent need to optimise prevention, diagnosis and treatment strategies [13]. Australia's Third National Hepatitis B Strategy (2018–2022) aims to reduce morbidity and mortality associated with CHB and sets clear targets to improve the cascade of care: 80% of the population to be diagnosed and aware of their infection; 50% of people living with CHB engaged in care; and 20% of people living with CHB to be on treatment [14]. Aboriginal and Torres Strait Islander people are considered a priority population in the strategy. Additionally, the strategy sets the goal of eliminating the negative impact of stigma and discrimination [14]. To address HBV-related mortality and to achieve elimination in Australia, substantial improvements in access to appropriate care, monitoring, and treatment are still required [6].

## Mobilising primary health care to increase access, equity, and improve health outcomes

In high CHB prevalence areas, such as the NT, the main route of HBV transmission is perinatal exposure at birth or in early childhood [15]. Given the life-long nature of CHB, there needs to be a shift away from specialist care and into a primary health care (PHC) model [16]. The Declaration of the Alma Ata, in 1978, provided the guiding foundation for PHC, stating that equitable, community-based health care are essential components to achieving "health for all" [17], with global recognition that community-based health workers are central to PHC success [18–20].

To achieve equitable health care and improve health outcomes for Aboriginal and Torres Strait Islander people, a health system needs to have culturally safe and accessible services for people on country [21,22]. The term "country" is used by Aboriginal and Torres Strait Islander people to describe a sense of belonging to land, a deeply spiritual relationship formed in the footsteps of ancestors. Country goes beyond geography, it incorporates kinship, totem, dance, ceremony, stories, language, and culture [23]. Care on country may help to reduce the adverse emotional impact Aboriginal and Torres Strait Islander people can face when they are forced to have care outside their own country [24,25] Delivering CHB care on country, in a PHC model, with a trained and supported Aboriginal health workforce, is central to achieving improved outcomes by increasing access, improving health literacy, and allowing patients to be close to their cultural support systems [26].

## Health literacy and chronic hepatitis B

Health literacy is an outcome of health education and communication, with roots in adult learning pedagogies and health promotion [27,28]. There is a well-recognised relationship between poor health literacy and poor health status [28]. A lack of culturally appropriate in-language resources and miscommunications between health staff and Aboriginal and Torres Strait Islander people is pervasive and a major barrier to achieving health and health literacy [29–32]. In a remote NT community, we found low levels of biomedical knowledge about CHB [33]. Low levels of CHB-specific health literacy and misconceptions around CHB have been identified in Aboriginal and Torres Strait Island people from other communities [34,35] and in populations throughout Australia where English is not their first language [36,37]. We have previously demonstrated that a shared understanding of CHB between patient and care provider is a pre-requisite to sustainable and quality engagement in care [32] and that health communication should be provided or reiterated in a patient's first language [33].

## Capacity building health workforce

Despite clear guidelines for the management of people living with CHB there is often limited implementation of best-practice care in a PHC space [34,35], which can be due to gaps in health provider knowledge and high staff turnover [35]. CHB-specific education exists for doctors and nurses. As there was no specific CHB-related education for the Aboriginal health workforce, we co-designed, developed and piloted the "Managing hepatitis B" training course using two-way learning and decolonising pedagogies [38], including Freirean pedagogy. Freirean pedagogy is a progressive educational approach rooted in the principles of critical thinking, social justice, and empowerment [39]. It involves praxis, an iterative cycle of awareness, and encompasses action with reflection and reflection with action [39]. The Aboriginal health workforce in the NT includes clinical and non-clinical roles. Aboriginal Health Practitioner (AHPs) have relevant qualifications in Aboriginal and Torres Strait Islander Primary Health Care practice, are professionally registered with Australian Health Practitioner Regulation Agency (AHPRA) and have clinical and specialist functions. Aboriginal Community Workers (ACWs) have a range of non-clinical roles, including health promotion. This study aims to evaluate our endorsed "Managing hepatitis B" training course for the Aboriginal health workforce and assess participants CHB knowledge, attitudes, and perception of skills.

# Materials and methods

## Study design, process, and context

This study is part of the Hep B PAST program is a partnership approach to sustainably eliminate CHB from the Aboriginal and Torres Strait Islander population of the NT [40]. This participatory action research (PAR) project aims to improve CHB-related health literacy, clinical care, and the cascade of care for people living with CHB [40]. A key component is building the capacity of primary health care professionals through the delivery of CHB education for doctors, nurses, and the Aboriginal health workforce.

**Study setting.**   The NT covers a large geographical area, 1.3 million km$^2$, and is sparsely populated, with a population of 233,000 [41]. Aboriginal and Torres Strait Islander people make up 26.3% of the population, with 77% living remote or very remotely [41]. Health care delivery in the NT is affected by remoteness, with 73% of communities >100km from the nearest hospital with no public transport and few household vehicles [42]. There are 87 primary health care centres throughout remote NT, mostly staffed by Aboriginal health workforce and resident nurses, supported by intermittent visits and telehealth from doctors and allied health staff [43,44].

**Researcher reflexivity.**   Our team has extensive experience in Aboriginal and Torres Strait Islander health, research, health education, primary health care and viral hepatitis. The Aboriginal and Torres Strait Islander members of the team come from diverse language and cultural groups throughout the NT. TDS, a Tiwi woman, has been an Aboriginal Health Practitioner (AHP) for 19 years. PMW, AHP and artist is a Ngangi speaker and lives in Nauiyu community on Malak Malak land. SW, a Warnidilyakwa woman and Anindilyakwa speaker from Groote Eylandt, has worked as an AHP for over 20 years. SN, a Gurindji woman and AHP for 24 years. GG is an Aboriginal Community Worker (ACW), researcher, proud Yolŋu man and Yolŋu Matha speaker. SB, TN, and AD are proud Yolŋu women, senior AHPs and Yolŋu Matha speakers. LB, an AHP for over 40 years, grew up in West Arnhem Land with Iwaidja speaking and mainland Kunwinku clans. CR is a proud Arrernte, Kaytete woman and has extensive family and kinship relationships across the NT. KS is a proud Arrernte and Anmatjere woman and senior AHP. The non-Indigenous members of the team (KH, PB, EVC, JTB, LW, RS, PS,

JSD, CC, JD) are critically conscious of the adverse impacts of colonisation [45]. We adhere to PAR and cultural safety principles [46,47], and continuously reflect and act to remove any potential biases [39,45,48].

## Recruitment of participants

The course was open to all Aboriginal health workforce in the NT. The course was advertised widely through networks and emails. We also used a targeted approach, with our team (KH, TDS, LB, SN) phoning health centres to encourage enrolment, informing health centre managers of the course content, benefits of participation and the role the Aboriginal health workforce can play in CHB care. The Aboriginal health workforce is a vital connection between community members and health services and act as a patient advocate. Their role in CHB care, management and prevention is varied and includes: screening - informing people of the need for a test and performing venepuncture; diagnosis - explaining to a person in their preferred language; management - explaining the importance of monitoring and care and supporting people to engage with the clinic; contact tracing–identifying family, kinship and household connections; vaccination; health promotion and reducing shame and stigma through improving CHB-related health literacy and education.

Recruitment into the study occurred at the beginning of the course and was open to all participants. Informed written consent was obtained.

## Ethical approval and consent to participate

Ethical approval was granted through the Human Research Ethics Committee of the Northern Territory Department of Health and Menzies School of Health Research - HREC 2018–3242 and Central Australia Human Research Ethics Committee of the Northern Territory - CAH-REC 18–3272 and Charles Darwin University Human Research Ethics Committee–H19008. This study was conducted in accordance with relevant guidelines and regulations including the Declaration of Helsinki and the NHMRC ethical conduct in research with Aboriginal and Torres Strait Islander Peoples and communities: Guidelines for Researchers and Stakeholders [47] and the Australian Institute of Aboriginal and Torres Strait Islander Studies Code of Ethics [49]. The authors had access to information that could identify individual participants. To protect participants all data, interviews and transcripts are treated as confidential and saved on secure drives, only accessible to researchers.

**Inclusivity in global research.** Additional information regarding the ethical, cultural, and scientific considerations specific to inclusivity in global research is included in S1 File.

## Course delivery and content

The one-and-a-half-day course was facilitated in English by Aboriginal and Torres Strait Islander (TDS, LB, SN, SW, PW, AD, TN, KS, GG, SB) and non-Indigenous health professionals (KH, JD, PB). The program is highly interactive, catering for varying literacy and professional levels. It includes presentations, case studies, role-playing, sharing stories and games. The course topics include the role of the Aboriginal health workforce in CHB management, care and support, epidemiology, transmission, prevention, liver anatomy and physiology and health promotion. We used the Hep B story app [33], which was developed to improve CHB-specific health literacy among Aboriginal and Torres Strait Islander communities by providing culturally appropriate education in a person's first language [33], currently available in nine NT Aboriginal languages [40]. The program has been designed to cater to the learning needs of the Aboriginal health workforce, modifying and adapting content considering context,

group dynamics, cultural requirements, knowledge, experience, and capacity of the participants [38,50].

## Course evaluation

We evaluated the course and its utility using selected quantitative and qualitative methodologies and evaluation tools specifically designed for this purpose [38].

**Quantitative evaluation.** We quantitatively measured course success, assessing CHB-related knowledge, attitude, and perception through questionnaires. We administered the questionnaires directly before the course, immediately after the course and three to six months after attending the course. We emailed follow up questionnaires or provided them to participants on outreach community visits (in 2019 and 2022 only due to COVID-19 travel restrictions).

**Statistical analysis.** We undertook statistical analysis using STATA 15.1. Differences across time points between pairs were tested using the Wilcoxon signed-rank test. Differences were considered significant at a $p < 0.05$.

**Qualitative data collection and analysis.** Consenting participants partook in semi-structured interviews. A mix of facilitators and research staff conducted interviews. We conducted and recorded interviews directly after the training and transcribed verbatim. Participants who did not want to be interviewed were offered the option to write answers to the semi-structured interview questions and/or join group discussions. Analysis was conducted aligning with PAR principles. Using grounded theory, we reviewed transcript recordings, written responses, and reflective journaling documents, allowing immersion in the data until data saturation point. We analysed the data inductively and deductively. We established and reviewed a coding-structure, building patterns, as categories and themes emerged. Data was coded in NVivo 12. A combination of Aboriginal and Torres Strait Islander and non-Indigenous team members (KH, EVC, LB, TDS, SW) came together several times and at distinct time points, April 2020, and March 2023, to review data interpretations and present findings and observations back to the group. We analysed the data collectively and agreed upon themes. Follow up data collection occurred between three and six months, through semi-structured interviews or written responses to the questions. Anecdotes, observations, and stories were provided by clinicians working with participants, and used with permission.

We deductively grouped codes into four categories when integrating and analysing the two forms of data; Level 1: reaction (satisfaction, engagement); Level 2: learning (knowledge and skills obtained); Level 3: behaviour (application of knowledge and skills) and Level 4: results (benefit to patients and observable results) [51]. These are the four domains described in Kirkpatrick's training evaluation framework for assessing pedagogical outcomes [51] and provides a structure to methodically describe changes in knowledge, attitude, perception and practice.

## Results

Eight "Managing hepatitis B" training courses for the Aboriginal health workforce were delivered between August 2019 and March 2023. The courses were delivered in four regional hubs in the NT, to increase access and equity to training opportunities. A total of 130 people attended from 32 different communities. See Fig 1 for a map of training locations and communities that participants work in. The rich language diversity of the NT was reflected with over 20 distinct first languages spoken by the participants, including Burarra, Kunwinjku, Ndjebbana, Yolu Matha, Tiwi, Ngangi, Anindilyakwa, Gurindji, Murrinh Patha, Arrernte, Kriol, Pitjantjatjara, Walpiri, Kaytetye, Nunggubuyu, Luritja, Warumungu, Anmatyerr and English.

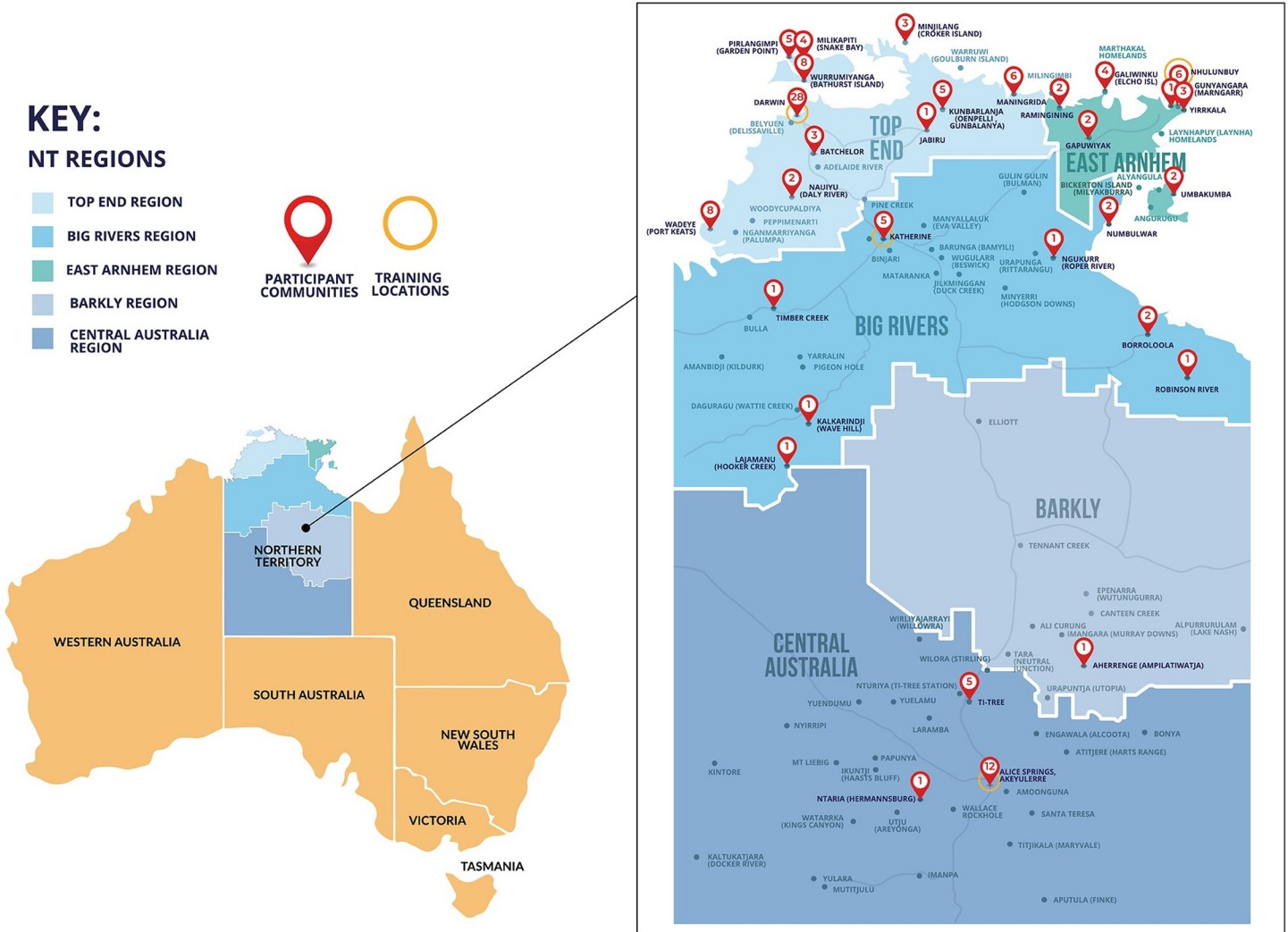

**Fig 1. Map of training locations and participant's communities.**

A total of 120 participants, 92.3% of all participants, completed at least one questionnaire. The demographics of the participants who completed pre- and post-course questionnaires are described in Table 1 and are representative of the whole cohort of participants.

Of the 120 people who answered the questionnaires there were 93 (77.5%) matched pairs completing both the pre- and post-course questionnaires, see Table 2. Results demonstrate statistically significant improvements in all domains.

## Qualitative analysis

We interviewed thirteen participants from Course 1. Eleven participants in courses 1–3 who opted out of being interviewed, consented to completing the interview prompt questions as written responses. Semi-structured interviews were not offered in courses 4–8, due to iterative feedback on interview acceptability and preference for written responses. Eighty-six participants responded to free text questions. A total of 110 individuals provided responses to contribute to thematic analysis, see Fig 2.

**Table 1. Demographics of the participants who completed pre- and post-course questionnaires.**

| Demographics | | Number of Participants | | | | |
|---|---|---|---|---|---|---|
| | | Total | Pre | Post | Paired Pre/Post | Follow up |
| | | n = 120 | n = 106 | n = 103 | n = 93 | n = 20 |
| Role | AHP | 64% | 58% | 55% | 51% | 11% |
| | ACW | 11% | 8% | 8% | 6% | 1% |
| | Other | 25% | 23% | 23% | 21% | 5% |
| Course | 1. Darwin 2019 (n = 26) | 22% | 18% | 17% | 17% | 8% |
| | 2. Katherine 2019 (n = 6) | 5% | 5% | 4% | 4% | 1% |
| | 3. Nhulunbuy 2019 (n = 9) | 8% | 7% | 7% | 6% | 3% |
| | 4. Darwin 2020 (n = 32) | 25% | 21% | 21% | 17% | 1% |
| | 5. Alice Springs 2021 (n = 14) | 12% | 8% | 10% | 7% | 1% |
| | 6. Darwin 2022 (n = 21) | 11% | 11% | 11% | 11% | 0% |
| | 7. Darwin 2023 (n = 12) | 10% | 10% | 10% | 10% | 3% |
| | 8. Alice Springs 2023 (n = 10) | 8% | 8% | 7% | 7% | 0% |
| Gender | Female | 77% | 68% | 68% | 62% | 14% |
| | Male | 21% | 19% | 18% | 15% | 3% |
| | Undisclosed | 2% | 2% | 1% | 1% | 0% |
| Age Group | 15–29 | 18% | 15% | 15% | 13% | 1% |
| | 30–39 | 25% | 22% | 19% | 17% | 5% |
| | 40–49 | 16% | 21% | 21% | 20% | 5% |
| | 50+ | 36% | 31% | 31% | 38% | 4% |
| | Undisclosed | 2% | 0% | 0% | 0% | 3% |

**Course acceptability and cultural safety.** Evaluations and feedback were overwhelmingly positive around the course content, delivery, methods, and facilitators. One participant stated:

> "I enjoyed how it was presented, everything was awesome. I really learned a lot    You's are AWESOME mob". (Course 2, written comment).

**Table 2. Pre- and post-course questionnaire results from 8 courses, % correct.**

| Competency question (correct answer) | | All | | | Paired Pre/Post | | |
|---|---|---|---|---|---|---|---|
| | | Pre | Post | Follow up | Pre | Post | p |
| | | (n = 106) | (n = 103) | (n = 20) | (n = 93) | | |
| Knowledge | Is hepatitis B common in Aboriginal and Torres Strait Islander people in the NT? (yes) | 84% | 98% | 100% | 85% | 98% | <0.001 |
| | Is hepatitis B a problem of the liver? (yes) | 88% | 96% | 100% | 89% | 97% | |
| | Is hepatitis B caused by alcohol? (no) | 35% | 61% | 80% | 33% | 62% | |
| | Is the main way people get hepatitis B in early childhood? (yes) | 44% | 98% | 95% | 47% | 98% | |
| | Can hepatitis B lead to liver cancer? (yes) | 76% | 93% | 95% | 76% | 94% | |
| Attitude | Is it a person's own fault if they have hepatitis B? (no) | 71% | 91% | 100% | 73% | 90% | <0.001 |
| | Is it safe for a person with hepatitis B to share meals with their family? (yes) | 57% | 96% | 100% | 56% | 97% | |
| Perception | Do you feel confident talking to patients, family, or community about hep B? (yes) | 61% | 95% | 100% | 63% | 95% | <0.001 |
| | Do you know how to use the Hep B Story app? (yes) | 29% | 93% | 100% | 29% | 95% | |
| | Do you know what you'd tell someone about keeping their liver healthy? (yes) | 65% | 97% | 100% | 68% | 97% | |
| | Do you know how to explain the main ways to get hepatitis B? (yes) | 57% | 97% | 100% | 58% | 97% | |
| **Overall** | | **61%** | **92%** | **97%** | **62%** | **93%** | |

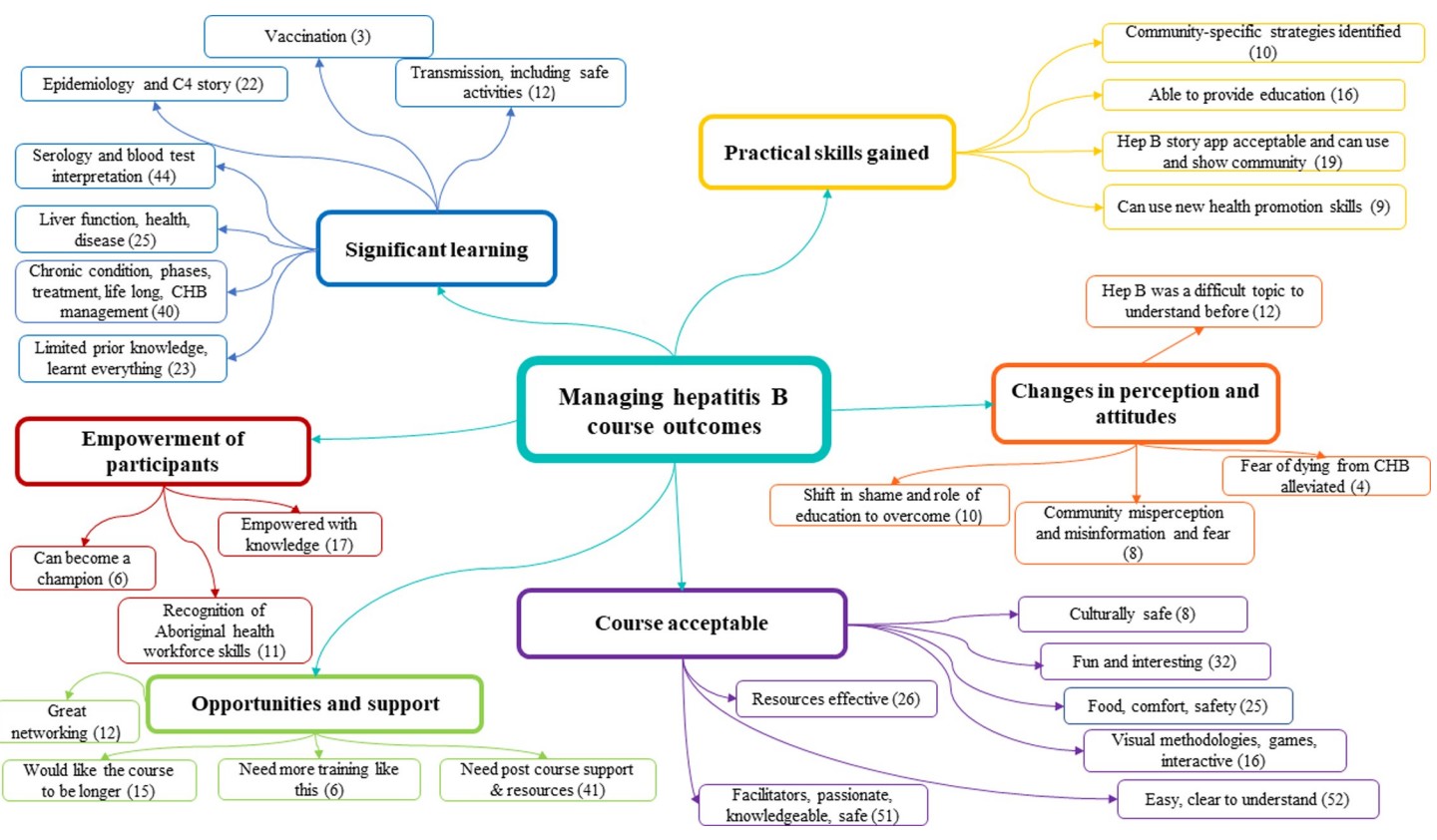

**Fig 2. Thematic network diagram representing overarching themes and subthemes (number of mentions).**

Ineffective communication in NT health settings has previously been reported [30,32,52]. Effectively conveying key messages clearly was crucial. Our findings demonstrate the course concepts were clear and easy to comprehend. Participants highlighted several factors that contributed to the success of the course, including delivery tailored to participant learning needs, support through relevant resources, and delivering content in a way that simplified and recapped key messages. A senior AHP, described that CHB can be a difficult subject but if delivered in a digestible manner, it can be easily understood:

> "It [the course] was really good. . .you did a really awesome job and I think you made it really in good language that everyone can understand–in simple language. Well, not simple but easy to understand 'cause Hepatitis B is a hard subject and for me". (Course 1, interview).

Positioning education using a cultural lens is always essential, and with CHB, there are additional, important sensitivities to consider in the NT context. HBV is transmitted through blood and body fluids,–which may include cultural ceremony and topics that can be considered as men's or women's business. The facilitator's consideration of this was acknowledged, with one participant describing the non-Indigenous presenter:

> "She's very mindful of you know cultural boundaries and she's constantly revising her presentation and approach". (Course 1, interview).

Throughout all courses, two-way learning and iterative changes were implemented to improve content and safety. One session focused on celebrating language diversity and learnt about the word and concepts of "liver" in different languages. In course 7 however, a participant shared with the group, *"we are stolen generation, our language was lost"*. The Stolen Generations refers to government laws, policies and practices which resulted in the forced removal of Aboriginal and Torres Strait Islander children from their families [53]. As facilitators we reflected on this valuable feedback and subsequently changed our approach when delivering this session, acknowledging the loss of language for some people due to colonisation and past racist policy and the dislocation and trauma caused for the Stolen Generations.

**Significant learning and practical skills gained.** The participants highlighted numerous knowledge and skill acquisition areas, including HBV modes of transmission, HBV serology interpretation, CHB management and treatment as a chronic condition, and CHB epidemiology. Participants were pleased to have CHB framed in a global perspective, with one participant commenting:

*'I think the statistical data on percentages representing Aboriginal and Torres Strait Islanders in the NT compared to the national level, worldwide was helpful, and the fact that the NT is kinda leading the way in regard to hep B prevention'.* (Course 1, interview).

And another.

*"It's [CHB] not just an Aboriginal problem".* (Course 3, written response).

The story of the C4 genotype and the long history of CHB in Aboriginal and Torres Strait Islander people in the NT [54] resonated with participants, with one commenting:

*"It's been in Aboriginal mob a long time this C4."* (Course 4, written response).

Participants gained a clearer understanding of CHB as a chronic condition, requiring life-long monitoring and care, rather than just seeing it as an infectious disease. With one participant commenting:

*"I was really surprised at how many Indigenous people actually do have hepatitis B and I didn't actually know that it really is an actual chronic condition that you can't cure".* (Course 1, interview).

And:

*"The downside of the infection if it's not treated. Like if you're not treated, it could lead to liver cancer."* (Course 4, written response).

Participants described how they would apply knowledge in their work and communities, demonstrating transferability of knowledge into practice. An example of how they would share learning with community, included:

*"[I will] share information I have learnt in morning meetings. Encourage all clinicians to routinely screen hep b status if unknown".* (Course 4, written response).

**Changes in perception and attitudes.** Shame was a theme frequently raised amongst our participants in semi-structured interviews. One participant mentioned *"it's a big shame job"*. The word *shame* is used frequently in many contexts by English speaking Aboriginal and Torres Strait Islander people in the NT. The concept of *shame* is complex and can extend beyond stigma and include embarrassment, shyness, discomfort, disrespect, be tied to social and spiritual obligations and respect for cultural responsibilities [55,56]. Misinformation around the HBV transmission was a theme that clearly emerged from the data, as well as the shame of it being associated with sexually transmissible infections (STIs), and fears of spreading and contracting HBV. Some participants had associated HBV only with injecting drug use or through occupational exposure. One participant stated:

*"I heard about Hepatitis B. . .know how the transferring of needles and learning about it in work as well because of you know handling needles and sharps and stuff like that. Yeah, before that, I didn't actually know that it could be transferred from mother to baby."* (Course 1, interview).

Education about transmission and clearly articulating how HBV is, and importantly is not transmitted and communicating safe activities, was identified as essential to reduce shame. Normalisation of HBV testing and discussions of CHB in communities were key strategies that emerged as a way of overcoming the shame and stigma associated with CHB. Encouraging the Aboriginal health workforce to openly discuss CHB was seen as an effective way to overcome shame. One participant said,

*"Lots of people don't know about hepatitis B, discussing hepatitis B with patients might help them a bit to understand that it's not a shame job. And maybe for the health workers to feel there's no shame about talking about it too.'* (Course 1, interview).

Participants highlighted the importance of delivering messages in-language. The Hep B story app was universally accepted, with participants expressing excitement to use in their own languages.

*'The app that's coming out I think that's a good resource. I'm really excited for the Kunwinjku one because that's what we'd be using. . .more resources like that, especially in-language, would be good.'* (Course 1, interview).

**Empowerment of participants.** Participants repeatedly raised the importance of being able to confidently explain CHB with patients. One participant commented that by providing this course it is:

*"Putting power back into the community".* (Course 1, interview).

Community education, in-language, was seen as a key to empower individuals to be active in their own CHB management:

*"You're empowering individuals to have you know, the correct information and then how to protect their family. It's just empowering people to have the information that allows them to teach their family members."* (Course 1, interview).

We found a clear thirst for more educational opportunities and a feeling of undervaluing the Aboriginal health workforce role. One participant describes:

*"There are not many opportunities for AHPs to have education like this [course]. . .nurses and doctors they are allowed, coming into Darwin for training. . .but not often for Aboriginal staff. . .we can learn more than they think. . .".* (Course 1, group discussion).

Our research clearly identifies that the Aboriginal health workforce are experts in their communities, with participants frequently raising strategies of how to implement knowledge and skills in their community. One participant mentioned:

*"With knowledge and understanding I can break it down into my own language   and go back and deliver that."* (Course 1, interview).

Having Aboriginal health workforce facilitators was also highlighted as a key asset. Positive role modelling and community empowerment was a recurring theme in the data, and having local facilitators supported this value. As one participant said:

*"I reckon that was really good how you did that [have AHP facilitators name]. . . .Yeah to see that, and sort of like leaders, gives other people more confidence. . . So, I think that's a really good thing."* (Course 1, interview).

Further, describing the vital role that the Aboriginal health workforce can play in ensuring patients understand diagnosis, treatment, and management:

*'This doctor thing, you know the language; how the doctor uses that big language. And even though we have our own interpreters back in the community and the family members interpreting, they still don't know, like us, Aboriginal health practitioners.'* (Course 1, interview).

**Opportunities and support.**   It was highlighted that clear simple messages that could be taken back into the community were important. The need for support after attending the course and a concern about retaining information was raised. Resources, information, teleconferences, and refresher courses were identified as strategies to ensure knowledge was maintained post-course. Supporting the community through resources such as the "Hep B Story" App, brochures and pamphlets, and physical resources such as model livers and electronic devices were seen as ways to share knowledge. The networking opportunities with colleagues from communities around the NT which was enabled by this course was highly valued, with strong calls for more training opportunities like this which bring together the Aboriginal health workforce.

**Follow up interviews and support.**   Fifty-six participants received a post course questionnaire with 20 participants completing it. The results showed a mean score of 97% demonstrating sustained and improved knowledge from all participants, see Table 2. Two people completed the 3-month post course semi-structured interviews and a further nine completed written responses. The findings support sustained and deep learning and application of knowledge and skills in their work and community. Knowledge acquired was numerous and varied, often reflecting different roles, for example, one participant described learning, included:

*"(I learnt about) promoting vaccinations which is applicable to my role"*. (Course 3, written response).

And:

"*Learning about Hep B and that it is a disease that affects the liver and how common it is in the NT. I didn't realise it can live in the body with little to no side effects, so that was also good to learn about. I also didn't realise that people with Hep B also have a higher chance of getting liver cancer. . .you need to get a blood test to know your status*". (Course 2, written response).

Another describes:

"*When I talked to my friends about it, it was good to also explain to them how important it is to keep our liver healthy. I think it also opened up a discussion to not only discuss Hep B but to also remember that drinking too much alcohol and having a poor diet can also affect the liver, so that was pretty good   also I show them the app*". (Course 1, interview).

HBV transmission was identified as a key learning, and by sharing this knowledge shame and fear can be shifted. One participant commented:

*"How it's passed on and like you have lots of people who think you get it from sharing food or cutleries, plates, like that you know and that was something I've learnt myself too, so I can explain it to people."* (Course 1, interview)

And another:

"*(I have) new information about Hep B and responding to the stigma around Hep B   Changing attitudes/behaviours & perhaps challenging peoples mind sets of shame and disappointments, around support how to engage within your family*". (Course 1, written response).

The importance of having training opportunities and showcasing the work and value of the Aboriginal health workforce continued to be a strong theme in the follow up interviews, with one participant reflecting:

*"I remember everything [from the course] but mostly how passionate some of the AHPs were and that they were now inspired (or re-inspired) to go back to their communities and share this critical information"*. (Course 2, written response).

And another:

*"That there are very strong workers in community doing amazing work and that these gatherings are important for them"*. (Course 1, written response).

Eight past participants have been mentored to teach on subsequent courses and five have been invited as authors (PW, TN, AD, SW, KS), building capacity in non-clinical areas and contributing to validating the findings in this paper. They have also presented this work at national and international conferences [40,57,58].

### Mixed methods evaluation of the course

Reviewing both the quantitative and qualitative data, using Kirkpatrick's model for assessing educational outcomes, demonstrates achievement of the aims of the study, see Table 3.

## Discussion

### Acceptability and cultural safety

The "Managing hepatitis B" training course for the Aboriginal health workforce led to significant CHB-related knowledge, and attitudinal and behavioural change. The study found that the course was culturally safe and acceptable to participants. PAR and culturally safe principles [45,46,60,61], responsive two-way learning, and Freirean pedagogy [39] contributed to course acceptability. Participant reflections and feedback facilitated critical consciousness within our team, which led to iterative changes to course content. To optimise learning for participants the course was developed to align with adult learning principles and resources [62] and Aboriginal and Torres Strait Islander pedagogical frameworks [63,64]. The course used a variety of methods to cater to learning styles including visual (PowerPoint slides), interactive (games) social (case study discussions) and kinaesthetic (role plays). The findings indicated that these methods were successful and catered to varying literacy and professional levels. English was not the first language of most participants. Despite this often being cited as a barrier to achieving learning outcomes our finding demonstrated that with culturally safe and decolonising methodologies significant learning can occur and, as previously found in the NT, using interactive, visual material, allowing space for stories and scenarios resulted in high attendance and participation, including for people with English as a second or third language [65].

We respectfully acknowledged difference, allowing flexibility for the participants and Aboriginal and Torres Strait Islander facilitators, including ensuring family and kinship needs were met so that the participants were able to fully engage in learning. We have previously described that before one can be empowered to learn they need to feel safe, secure and sustained [38]. The importance of these factors was emphasised by participants with non-content related themes emerging in the data. These included suggestions such as controlling noise and light levels, satisfaction with the supply of breakfast, the course being held at suitable (or not, in Course 4) locations, enjoying the food provided and feeling safe and respected.

Given the high CHB prevalence in the NT, we were acutely aware of the potential that participants could be living with or affected by CHB. This was acknowledged, with an open offer for participants to privately talk to facilitators anytime during or after the course if content raised questions or concerns. Several people took the opportunity to seek clarification, get advice and shared stories of family loss to HCC and their own personal experiences. One participant living with CHB spoke to a facilitator privately, after a session had caused worry that they could never have an alcoholic beverage again. This person was counselled around the risks in varying CHB phases and reassured. They were thanked for this valuable feedback, and guaranteed this session would be clearer and more nuanced for future courses. The psychological impact experienced by people living with or affected by CHB, including fear of disease progression, infecting others and internal stigma have previously been described [66]. Similarly, our data reflected this, with people identifying they now no longer feared that CHB was a "death sentence" for them.

### Primary health care and health literacy

Differences between Aboriginal and Torres Strait Islander and biomedical concepts of health, culture and worldview are factors that can influence achieving health literacy [32,67].

**Table 3. Study results measured using Kirkpatrick's model [51].**

| Kirkpatrick model level | How it was measured | Sample size | Evidence of achievement |
|---|---|---|---|
| **Level 1: Reactions**<br>How did the participants feel about the course? | Semi-structures interviews and free text questions | 110 | 210 mentions about course acceptability, included content was clear and easy to understand, the course was fun and interesting, the facilitators were passionate, knowledgeable, and culturally safe, methodologies facilitated learning, i.e., visual, and interactive.<br>• *"Great presenters, communication, clear/concise,"* (Course 7)<br>• *"You were able to cater to all different levels and needs".* (Course 1)<br>• *"The passion and knowledge of the facilitators captured my attention all day"* (Course 7)<br>• *"Group work, scenarios and case studies. Allowed everyone to get involved".* (Course 6) |
| **Level 2: Learning**<br>Did knowledge, attitude and perception of skills improve? | Pre and post-test questionnaires Semi-structures interviews and free text questions | 93 matched pairs 110 | 31% overall improvement. Statistically significant improvements, *p* <0.001, in all domains.Knowledge: (169 mentions in many different topic areas)<br>• *"I learnt so much. I didn't know a drop about hep B [before course]".* (Course 7)<br>• *"Passed on from mum to bub, early childhood and children can get it through wound transmission"* (Course 5).<br>• *"Understanding bloods, and follow up and treatment"* (Course 6)<br>• *"Been in Aboriginal mob a longtime"* (Course 7)<br>Attitude: (34 mentions of change in attitude)<br>• *"Hep B doesn't have to be a death sentence."* (Course 2)<br>• *"More education will let people know that it's not their fault and stuff like that . . .and you will get support".* (Course 3)<br>• *"I know now it's not a shame job".* (Course 1)<br>Perception of skills: (88 mentions of using new skills and knowledge, with practical examples)<br>• *"I have a better understanding of hep b I can help my people understand."* (Course 3)<br>• *"The hep B story app useful 'cause I found a lot of it on my phone and I've been looking at it in English. Then like I can interpret it or break it down in Kriol Pidgin English or my own language."* (Course 4) |
| **Level 3: Behaviour**<br>To what extent did participants change their behaviour back in their clinics or community? | 3-month post training questionnaire 3-month post training Semi-structures interviews and free text questions | 20 11 | Demonstrated sustained and improved knowledge. The overall score of 97% correct.Sustained learning, sharing of knowledge and change in practice articulated:<br>• *"For me it was these key factors, Hep B old information, about the status of Hep B/CHB, how common it is today. I am now helping people to know whether they have it or not. I know what screening is required for Hep B. C4 Ancient disease".* (Course 1)<br>• *"Hepatitis B is not just an NT/Australian disease that is affecting Aboriginal people. I remember that Aboriginal people are suffering from a specific genotype of hep B called C4".* (Course 2)<br>• *"The differences between Hep A, B and C and how they are transmitted"* (Course 7). |

*(Continued)*

**Table 3.** (Continued)

| Kirkpatrick model level | How it was measured | Sample size | Evidence of achievement |
|---|---|---|---|
| **Level 4: Results**<br>What impact or benefit has resulted from the training? | Clinic data on cascade of care 3-month post training Semi-structures interviews and free text questions<br>Anecdotes, observations, success stories | | **National targets:** 80% aware of infection; 50% engaged in care; 20% on treatment.<br>**NT data pre Hep B PAST, 2018 [59]:** 73.8% aware of infection; 27.8% engaged in care; 7.2% on treatment.<br>**Trained Aboriginal workforce & Hep B PAST 2022 [40]:** 92% aware of infection; 70% engaged in care; 22% on treatment.<br> • *"I told my family about hep B and they need to get a test to know their status."* (Course 1)<br> • *"I talked to Jenny (manager) about getting more information and how I can help people within my community with Hep B"* (Course 3)<br>*"He has come back a changed person, he seemed really disengaged before but now he has a purpose. He has found a person we've been trying to find for ages, got him into the clinic to have a test and (patient) now has a hep B diagnosis and is in care"* (AHP coordinator describing participant post training).<br>*"He was able to explain (in patients first language) why treatment was needed and now they've (patient) started on tenofovir. . . .we'd been trying for years"*. (Nurse Practitioner (NP) supporting ACW post training).<br>*"They have got everyone on the list to come to clinic, they do so much groundwork! They have helped to Fibroscan, take bloods and explaining care using the app"* (Liver NP describing past participants assistance on liver clinic outreach visit)<br>*"She has now recalled everyone in her community. They now have 99% of the community with a Hep B status on their problem list. They are smashing the National targets"* (Hep B Program manager describing past participants efforts to serocode community)<br>*"He was able to see family relationships that we were not aware of. He has a clear understanding of transmission and contact tracing and we have subsequently been able to diagnose new people and put them on the correct care pathway"*. (Specialist on liver clinic outreach visit, supported by a past AHP participant). |

Balancing health concepts was a priority, so that shared understandings could be reached between facilitators and participants, and participants and their patients. Keeping information simple yet informative was affirmed by participants' responses indicating information was clear and easy to understand, even though CHB had traditionally been a difficult area of health to learn. This is consistent with other studies, finding CHB is often misunderstood by those living with CHB, their families, and communities, and those involved in their care and management [32,35,68]. PHC practitioners including doctors, nurses and the Aboriginal health workforce play a central role in identifying, testing, and managing CHB. However, there are often significant gaps in their CHB-related knowledge, particularly regarding ordering and interpreting tests and best practice guidelines for managing HBV as a chronic condition [69]. Previous difficulty with serological interpretation and a new understanding of CHB as a life-long condition emerged from the data as key areas of learning.

Other reported barriers to providing guideline-based care in the NT include modifying information based on the clinician's perceptions of the patient's level of English, limited consultation time, doctors and nurses in remote clinics having high workloads and competing demands [70]. The Aboriginal health workforce is an important resource to overcome these barriers and are pivotal in providing care to people living with CHB and in educating and empowering community members. A strong theme of empowerment through knowledge, for both individuals and community, was evident in the data.

## Transmission, misconceptions, and shame

Misconceptions around modes of transmission, a lack of understanding surrounding the need for regular care and the risks of unmonitored CHB have been observed [32,35,71]. We purposefully focused on transmission, myth busting, reframing CHB as a chronic condition [34], and supporting participants to reflect on long-standing beliefs around CHB. Our findings demonstrated strong learning around transmission and revealed previous misunderstandings about modes of HBV transmission, including sharing meals and cutlery. This misconception that HBV can be transmitted through sharing food has been found in other studies [37].

Stigma associated with chronic infectious diseases is well characterised and is known to be a major barrier to people seeking testing, monitoring or treatment [71]. Despite a much higher global prevalence and mortality than HIV [1], stigma associated with CHB is less well defined [71], partly due a lower global prioritisation. However, it is recognised that people living with CHB experience stigma in many forms [72]. HBV-related stigma has been found to be related to knowledge deficits and fear [71–73]. We found some participants described shame in talking about HBV and a general lack of HBV health literacy and misinformation regarding routes of transmission amongst health care workers and community. Participants associated HBV with STIs and injecting drug use. Studies have suggested that stigma may be higher in low HBV prevalence areas, where the majority of HBV transmission is through sex or injecting drug use, further marginalising, and stigmatising people in these groups due to their perceived risk of HBV infection [73]. However, in Aboriginal and Torres Strait Islander people in the NT, CHB is endemic, with most transmission occurring in early childhood, yet we found that fear and shame to be strongly present in our study.

We are undertaking further research to identify and understand CHB shame in our context and to identify and evaluate interventions to effectively address shame and stigma. Presently, the participants have suggested strategies to help overcome shame, largely around education and information. They discussed the importance of normalising discussions about CHB in the community to overcome shame. Focusing on transmission, including what is safe and prevention messages around vaccination and antenatal care were identified as effective strategies.

Best practice CHB care involves management from a variety of health care professionals. Having the Aboriginal health workforce at the centre of this PHC model facilitates optimal care for the patient and can assist as a mediator for other health professionals. As evident in our follow up data we found that the Aboriginal health workforce can fill gaps in patient education and encourage adherence to monitoring and treatment as they can explain concepts in a way that is appropriate to patients. We found the Aboriginal health workforce play an important role in educating other members of the health workforce, their own families, and communities, improving CHB understanding and care.

## Limitations and challenges

Having Aboriginal and Torres Strait Islander facilitators was important to the participants, as role models and to re-invigorate enthusiasm for their professions. The co-design process included a focus to capacity build facilitators in CHB knowledge and teaching methodologies [38]. There is a sense of ownership, trust and respect that shone through and may have been a component to the success of the course, with the passion of facilitators mentioned by participants. For sustainability and successful transferability of this course to be achieved there must be an Aboriginal and Torres Strait Islander facilitator. All facilitators should adhere to principles of culturally safe training, be trained and skilled in CHB and teaching methodologies [38].

Post training support, mentoring and recaps of keys messages help to consolidate skills and knowledge [65]. We have provided ongoing support through facilitators, who are AHP

coordinators in the regions, provided opportunities, including working with the liver clinic when visiting community, and linked participants with doctors and nurses in their area who have undergone equivalent training, where possible. However, we were unable to provide all participants with face-to face catch ups and mentoring due to the vast geography of the NT. COVID-19 travel restrictions meant remote clinic visits during 2020–2022 ceased and subsequent staff shortages and burnout have made follow-up education in communities more challenging.

The semi-structured interviews were conducted by Aboriginal and Torres Strait Islander and non-Indigenous facilitators and research staff. We acknowledge that having facilitators as interviewers may have influenced the responses given. Conversely, our Aboriginal and Torres Strait Islander research and teaching team also thought this might act as an enabler as *"we may get more accurate answers when there is connection and trust"*. We observed measurable improvements in the cascade of care data in clinics with a trained Aboriginal health workforce member, aligning with level 4 of Kirkpatrick's framework, observable benefits and impacts on organisations. However, we agree with Kirkpatrick and others who state that it is difficult to evaluate the direct impact of training at this level, due to the inability to separate other factors and interventions that may also be contributing to improved results [38,74]. This finding will be further explored in the overall analysis of the entire Hep B PAST intervention.

## Conclusion

Through our mixed method evaluation, we can confidently validate that the "Managing hepatitis B" training course for the Aboriginal health workforce is contributing to significant improvements in knowledge. Themes relating to course acceptability, cultural safety, shame and previous misinformation and misconceptions about transmission were repeatedly identified. Strong acquisition of knowledge in all key topic areas and examples of how to apply knowledge and skills in community were well articulated by participants. We observed great enthusiasm and a strong desire for more educational opportunities to improve health outcomes. Our training supported the Aboriginal health workforce to be at the centre of care provision. This has created a network of support and opportunity for ongoing partnership, mentoring and training, positively contributing to sustained knowledge and commitment to CHB care.

## Supporting information

**S1 File. Inclusivity in global research.**
(DOCX)

**S2 File. Hep B PAST partnership members.**
(PDF)

## Acknowledgments

We would like to thank all the participants of this course. We would also like to thank their managers for supporting attendance. We acknowledge the Hep B PAST partnership members S2 File. Thank you to Melita McKinnon, Tammy-Allyn Fernandes, Mikaela Mobsby, Tiana Alley, Beatrice Parry, Timothy Nabegeyo, Belinda Garling, Natasha Tatipata and Rafael Espinoza-Hosking for providing administrative and other support. Thank you to Nurse Practitioners Matthew Maddison and Stuart Mobsby for providing mentoring and support. Aboriginal and Torres Strait Islander peoples are the first peoples of Australia and have a strong and resilient history. There is no single Aboriginal and Torres Strait Islander culture or group. The

authors acknowledge and pay respect to the past and present Traditional Custodians and Elders, and the continuation of cultural, spiritual, and educational practices of all Aboriginal and Torres Strait Islander peoples. Throughout this paper the Aboriginal and Torres Strait Islander health workforce are referred to as "Aboriginal health workforce" in-line with the language used for these professions in the Northern Territory. The authors acknowledge the important role the of the Aboriginal health workforce.

## Author Contributions

**Conceptualization:** Kelly Hosking, Christine Connors, Jane Davies.

**Data curation:** Kelly Hosking, Linda Ward.

**Formal analysis:** Kelly Hosking, Teresa De Santis, Emily Vintour-Cesar, Phillip Merrdi Wilson, Linda Bunn, Shiraline Wurrawilya, Sandra Nelson, Cheryl Ross, Linda Ward.

**Funding acquisition:** Joshua S. Davis, Christine Connors, Jane Davies.

**Investigation:** Kelly Hosking, Teresa De Santis, Emily Vintour-Cesar, Phillip Merrdi Wilson, Linda Bunn, George Garambaka Gurruwiwi, Shiraline Wurrawilya, Sarah Mariyalawuy Bukulatjpi, Sandra Nelson, Cheryl Ross, Kelly-Anne Stuart-Carter, Terese Ngurruwuthun, Amanda Dhagapan, Paula Binks, Richard Sullivan, Jaclyn Tate-Baker.

**Methodology:** Kelly Hosking, Teresa De Santis, Phillip Merrdi Wilson, Linda Bunn, George Garambaka Gurruwiwi, Shiraline Wurrawilya, Sarah Mariyalawuy Bukulatjpi, Sandra Nelson, Kelly-Anne Stuart-Carter, Terese Ngurruwuthun, Amanda Dhagapan, Paula Binks, Richard Sullivan, Phoebe Schroder, Jaclyn Tate-Baker, Joshua S. Davis, Jane Davies.

**Project administration:** Kelly Hosking, Emily Vintour-Cesar, Cheryl Ross, Paula Binks.

**Resources:** Kelly Hosking, Teresa De Santis, Phillip Merrdi Wilson, Linda Bunn, Shiraline Wurrawilya, Phoebe Schroder.

**Supervision:** Joshua S. Davis, Christine Connors, Jane Davies.

**Validation:** Phillip Merrdi Wilson, Shiraline Wurrawilya.

**Visualization:** Kelly Hosking, Teresa De Santis, Linda Bunn.

**Writing – original draft:** Kelly Hosking.

**Writing – review & editing:** Kelly Hosking, Teresa De Santis, Emily Vintour-Cesar, Phillip Merrdi Wilson, Linda Bunn, George Garambaka Gurruwiwi, Shiraline Wurrawilya, Sarah Mariyalawuy Bukulatjpi, Sandra Nelson, Cheryl Ross, Kelly-Anne Stuart-Carter, Terese Ngurruwuthun, Amanda Dhagapan, Paula Binks, Richard Sullivan, Linda Ward, Phoebe Schroder, Jaclyn Tate-Baker, Joshua S. Davis, Christine Connors, Jane Davies.

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
