## [Decision Letter · Decision Letter 0]

22 Nov 2023

PONE-D-23-20121“Putting the power back into community”: A mixed methods evaluation of a chronic hepatitis B training course for the Aboriginal health workforce of Australia’s Northern TerritoryPLOS ONE

Dear Dr. Hosking,

Thank you for submitting your manuscript to PLOS ONE. After careful consideration, we feel that it has merit but does not fully meet PLOS ONE’s publication criteria as it currently stands. Therefore, we invite you to submit a revised version of the manuscript that addresses the points raised during the review process.  The reviewers both found the paper to be highly readable and informative. I agree with them!  Reviewers' comments while minimal will improve this comprehensive paper. 

We look forward to receiving your revised manuscript.

Kind regards,

Kimberly Page, PhD, MPH

Academic Editor

PLOS ONE

Journal Requirements:

"This research is part of the Hep B PAST project, which receives a National Health and Medical Research Council (NHMRC) partnership grant, GNT1151837. https://www.nhmrc.gov.au/ 

KH is undertaking a PhD and has an NHRMC scholarship, GNT1190918."

7. One of the noted authors is a group or consortium [Hep B PAST partnership]. In addition to naming the author group, please list the individual authors and affiliations within this group in the acknowledgments section of your manuscript. Please also indicate clearly a lead author for this group along with a contact email address.

8. We note that Figure 1 in your submission contain map/satellite images which may be copyrighted. All PLOS content is published under the Creative Commons Attribution License (CC BY 4.0), which means that the manuscript, images, and Supporting Information files will be freely available online, and any third party is permitted to access, download, copy, distribute, and use these materials in any way, even commercially, with proper attribution. For these reasons, we cannot publish previously copyrighted maps or satellite images created using proprietary data, such as Google software (Google Maps, Street View, and Earth). For more information, see our copyright guidelines: http://journals.plos.org/plosone/s/licenses-and-copyright.

Reviewers' comments:

Reviewer's Responses to Questions

**Comments to the Author**

1. Is the manuscript technically sound, and do the data support the conclusions?

Reviewer #1: Yes

Reviewer #2: Yes

2. Has the statistical analysis been performed appropriately and rigorously? 

Reviewer #1: Yes

Reviewer #2: Yes

3. Have the authors made all data underlying the findings in their manuscript fully available?

Reviewer #1: Yes

Reviewer #2: Yes

4. Is the manuscript presented in an intelligible fashion and written in standard English?

Reviewer #1: Yes

Reviewer #2: Yes

5. Review Comments to the Author

Reviewer #1: Thank you for the opportunity to review this manuscript titled: “Putting the power back into community”: A mixed methods evaluation of a chronic hepatitis B training course for the Aboriginal health workforce of Australia’s Northern Territory. The authors must be congratulated on producing such a high quality manuscript that provides significant detail about the study, findings and context; the latter being incredibly important in teaching others how to conduct codesign methodologies successfully. I am extremely impressed by the writing, it is clear and without errors. I have only minor comments (which seems extraordinary considering that this is a long paper):

L143 - I realise you provide a description of the qualifications of the Aboriginal health workforce in the acknowledgements, but I think it would be useful to briefly describe the role, including qualification and scope of practice, in the body of the paper.

L182 - the scope of the Aboriginal health workforce in delivering CHB care needs to be described.

L231 - Who conducted the interviews after the training? Were the facilitators also the evaluators? This needs to be explained and acknowledged in the limitations or explained.

L243 - minor spelling mistake - "interviews" not "interviewed".

L556 - insert "a" - "...is known to be [a] major barrier..."

This manuscript is a joy to read because it is well written and generous in sharing all the insights and tips gathered through conducting such an extensive project. That you have demonstrated an improvement in knowledge and attitudes through delivery of this course and have expanded the CHB workforce in the Northern Territory is remarkable. Congratulations.

Reviewer #2: “Putting the power back into community”: A mixed methods evaluation of a chronic

hepatitis B training course for the Aboriginal health workforce of Australia’s Northern

Territory

Introduction

Clear, comprehensive

Might have been good to detail which Aboriginal languages were included in the team identity and the participants.

Would have been good to outline Freirian principles of pedagogy.

Methods

Appropriate

Would have been good in “Reflexivity” to mention specific cultural background was represented amongst the Aboriginal member of the team

Results

Clear, concise, accurate

Would have been good to list Aboriginal languages spoken amongst Aboriginal Liaison languages spoken.

Discussion

Very comprehensive

6. PLOS authors have the option to publish the peer review history of their article (what does this mean?). If published, this will include your full peer review and any attached files.

Reviewer #1: No

Reviewer #2: **Yes: **Christopher Numa Lemoh

---

## [Author Response · Author response to Decision Letter 0]

11 Dec 2023

Thank you for your feedback, it has improved our manuscript.

Reviewers feedback

Reviewer #1: Thank you for the opportunity to review this manuscript titled: “Putting the power back into community”: A mixed methods evaluation of a chronic hepatitis B training course for the Aboriginal health workforce of Australia’s Northern Territory. The authors must be congratulated on producing such a high quality manuscript that provides significant detail about the study, findings and context; the latter being incredibly important in teaching others how to conduct codesign methodologies successfully. I am extremely impressed by the writing, it is clear and without errors. I have only minor comments (which seems extraordinary considering that this is a long paper):

Thank you for your generous feedback.

L143 - I realise you provide a description of the qualifications of the Aboriginal health workforce in the acknowledgements, but I think it would be useful to briefly describe the role, including qualification and scope of practice, in the body of the paper.

Thank you. Yes, we agree that describing this in the introduction will help the reader. I have added the following.

“The Aboriginal health workforce in the NT includes clinical and non-clinical roles. Aboriginal Health Practitioners (AHPs) have relevant qualifications in Aboriginal and Torres Strait Islander Primary Health Care practice, are professionally registered with the Australian Health Practitioner Regulation Agency (AHPRA) and have clinical and specialist functions. Aboriginal Community Workers (ACWs) have a range of non-clinical roles, including health promotion.”

L182 - the scope of the Aboriginal health workforce in delivering CHB care needs to be described.

Thank you. I have added the following.

“The Aboriginal health workforce is a vital connection between community members and health services and act as a patient advocate. Their role in CHB care, management and prevention is varied and includes screening - informing people of the need for a test and performing venepuncture, diagnosis - explaining to a person in their preferred language, management - explaining the importance of monitoring and care and supporting people to come into clinic, contact tracing - understanding family, kinship and household connections, vaccination; health promotion and reducing shame and stigma through improving health literacy and education.”

L231 - Who conducted the interviews after the training? Were the facilitators also the evaluators? This needs to be explained and acknowledged in the limitations or explained.

Thank you, this is an important point. I have added this to the methods and to the discussion in the limitations.

"A mix of facilitators and research staff conducted interviews”. 

“The semi-structured interviews were conducted by Aboriginal and Torres Strait Islander and non-Indigenous facilitators and research staff. We acknowledge that having facilitators as interviewers may have influenced the responses given. Conversely, our Aboriginal and Torres Strait Islander research and teaching team also thought this might act as an enabler as “we may get more accurate answers when there is a connection and trust”.

L243 - minor spelling mistake - "interviews" not "interviewed".

Thank you. I have corrected it to “interviews”.

L556 - insert "a" - "...is known to be [a] major barrier..."

Thank you, “a” is now inserted.

This manuscript is a joy to read because it is well written and generous in sharing all the insights and tips gathered through conducting such an extensive project. That you have demonstrated an improvement in knowledge and attitudes through delivery of this course and have expanded the CHB workforce in the Northern Territory is remarkable. Congratulations.

Thank you, very much appreciated.

Reviewer #2: “Putting the power back into community”: A mixed methods evaluation of a chronic

hepatitis B training course for the Aboriginal health workforce of Australia’s Northern

Territory

Introduction

Clear, comprehensive

Might have been good to detail which Aboriginal languages were included in the team identity and the participants.

Would have been good to outline Freirian principles of pedagogy.

Thank you. 

I have now included the language and cultural groups of the Aboriginal and Torres Strait Islander teaching and research team in materials and methods, in researcher reflexivity. 

“TDS, a Tiwi woman, has been an Aboriginal Health Practitioner (AHP) for 19 years. PMW, AHP and artist is a Ngangi speaker and lives in Nauiyu community on Malak Malak land. SW, a Warnidilyakwa woman and Anindilyakwa speaker from Groote Eylandt has worked as an AHP for over 20 years. SN, a Gurindji woman and AHP for 24 years. GG is an Aboriginal Community Worker (ACW), researcher, proud Yolŋu man, and Yolŋu Matha speaker. SB, TN, and AD are proud Yolŋu women and senior AHPs and Yolŋu Matha speakers. LB, an AHP for over 40 years, grew up in West Arnhem Land with Iwaidja speaking and mainland Kunwinku clans. CR is a proud Arrernte, Kaytete woman and has extensive family and kinship relationships across the NT. KSC is a proud Arrernte and Anmatjere woman and senior AHP.”

I have added the participants languages in the results.

“The rich language diversity of the NT was reflected with over 20 distinct first languages spoken by the participants, including Burarra, Kunwinjku, Ndjebbana, Yolŋu Matha, Tiwi, Ngangi, Anindilyakwa, Gurindji, Murrinh Patha, Arrernte, Kriol, Pitjantjatjara, Walpiri, Kaytetye, Nunggubuyu, Luritja, Warumungu, Anmatyerr and English.”

I have now outlined Freirean pedagogy in the introduction with the following.

“Freirean pedagogy is a progressive educational approach rooted in the principles of critical thinking, social justice, and empowerment (ref). It involves praxis, an iterative cycle of awareness, and encompasses action with reflection and reflection with action (ref). 

Methods

Appropriate

Would have been good in “Reflexivity” to mention specific cultural background was represented amongst the Aboriginal member of the team

Thank you. We are delighted to add more detail. We had reduced due to the length of the manuscript. In consultation with the Aboriginal research and teaching team, we have included. 

“TDS, a Tiwi woman, has been an Aboriginal Health Practitioner (AHP) for 19 years. PMW, AHP and artist, is a Ngangi speaker and lives in Nauiyu community, on Malak Malak land. SW, a Warnidilyakwa woman and Anindilyakwa speaker from Groote Eylandt, has worked as an AHP for over 20 years. SN, a Gurindji woman and AHP for 24 years. GG is an Aboriginal Community Worker (ACW), researcher and proud Yolŋu man and Yolŋu Matha speaker. SB, TN, and AD are proud Yolŋu women and senior AHPs and Yolŋu Matha speakers. LB, an AHP for over 40 years, grew up in West Arnhem Land with Iwaidja speaking and mainland Kunwinku clans. CR is a is a proud Arrernte, Kaytete woman and has extensive family and kinship relationships across the NT. KS is a proud Arrernte and Anmatjere woman and senior AHP.”

Results

Clear, concise, accurate

Would have been good to list Aboriginal languages spoken amongst Aboriginal Liaison languages spoken.

Thank you for this suggestion. I have included the following in the results:

“The rich language diversity of the NT was reflected with over 20 distinct first languages spoken by the participants, including Burarra, Kunwinjku, Ndjebbana, Yolŋu Matha, Tiwi, Ngangi, Anindilyakwa, Gurindji, Murrinh Patha, Arrernte, Kriol, Pitjantjatjara, Walpiri, Kaytetye, Nunggubuyu, Luritja, Warumungu, Anmatyerr and English.”

Discussion

Very comprehensive

Thank you, very much appreciated.

Journal Requirements:

Thank you, I have made the necessary adjustments to the font, headings, and format.

Thank you. I have completed and uploaded the “Inclusivity in global research form “with the revised manuscript. 

I have included the following in the methods section.

Inclusivity in global research

“Additional information regarding the ethical, cultural, and scientific considerations specific to inclusivity in global research is included in the Supporting Information (S1 Checklist).”

Thank you. Our ethics does not allow this.

"This research is part of the Hep B PAST project, which receives a National Health and Medical Research Council (NHMRC) partnership grant, GNT1151837. https://www.nhmrc.gov.au/

KH is undertaking a PhD and has an NHRMC scholarship, GNT1190918."

Thank you for amending the Funding Statement in the online submission form. I have included the below in the cover letter.

“The NHMRC supported this work. This research is part of the Hep B PAST project, which receives an NHMRC partnership grant, GNT1151837. KH is undertaking a PhD and has an NHRMC scholarship, GNT1190918. Funders played no role in the study design, analysis or the decision to publish. There was no additional external funding received for this study.”

I have also added “There was no additional external funding received for this study” to the manuscript.

Thank you could you please amend our Data Availability statement to reflect that there are ethical restrictions to sharing our data publicly. Please see below, which I have added to the cover letter.

I have added the following to the cover letter.

“There are ethical restrictions to sharing our data publicly. The ethics approval specifies, “No third party will be given access to or copies of the data”, Human Research Ethics Committee of the Northern Territory Department of Health and Menzies School of Health Research - HREC 15-2417. There were additional privacy concerns, from within the Aboriginal and Torres Strait Islander author group and from the Aboriginal and Torres Strait Islander communities involved in Hep B PAST, of sharing de-identified data, as it was perceived that the potential for re-identification is greater given the small population and workforce size. Data may be available for reasonable requests through the Hep B PAST steering committee, email: Hepbpast@menzies.edu.au. Phone +61889468687 or kelly.hosking@menzies.edu.au Phone +61 472817647 as the corresponding author.

Thank you for updating the Data Availability statement to reflect the information provided in my cover letter.

Thank you, and I apologise, I must have incorrectly selected that option.

Please amend our Data Availability statement to reflect that there are ethical restrictions to sharing our data publicly. Please see above, which I have added to the cover letter. Please refer to the cover letter

Data may be available for reasonable requests through to the Hep B PAST steering committee, email: Hepbpast@menzies.edu.au. Phone +61889468687 or kelly.hosking@menzies.edu.au. 

7. One of the noted authors is a group or consortium [Hep B PAST partnership]. In addition to naming the author group, please list the individual authors and affiliations within this group in the acknowledgements section of your manuscript. Please also indicate clearly a lead author for this group along with a contact email address.

Thank you, I have added to the acknowledgements section of the manuscript. 

“We acknowledge the Hep B PAST partnership. See Supplementary Information (S2).”

 A/Prof Jane Davies is the Chief Investigator for Hep B PAST and can be contacted at jane.davies@menzies.edu.au. Kelly Hosking is this manuscript's lead and corresponding author and can be contacted at kelly.hosking@menzies.edu.au.

I have uploaded the members of the consortium/partnership within the manuscript as S2.

When re-reviewing the authorship requirement, I believe one of the people named as an acknowledgement, Cheryl Ross, meets the authorship criteria. I have added her to the authorship list. Can PLOS ONE please update the online submission to reflect this, if required? Cheryl has read the manuscript and consents to it being submitted/published. Her affiliation is Menzies School of Health Research. Her contact details are: Cheryl.ross@menzies.edu.au

8. We note that Figure 1 in your submission contain map/satellite images which may be copyrighted. All PLOS content is published under the Creative Commons Attribution License (CC BY 4.0), which means that the manuscript, images, and Supporting Information files will be freely available online, and any third party is permitted to access, download, copy, distribute, and use these materials in any way, even commercially, with proper attribution. For these reasons, we cannot publish previously copyrighted maps or satellite images created using proprietary data, such as Google software (Google Maps, Street View, and Earth). For more information, see our copyright guidelines: http://journals.plos.org/plosone/s/licenses-and-copyright.

This was created by NT Health communications and media team. We have discussed with the legal team and have the correct permission to publish. The “Request for Permission to Publish Content “has been signed and uploaded as “Other”.

One reference “38” has changed from “preprint” to published. 

Reviewer 2 suggested adding a description of Freirean pedagogy to the introduction, which I have done. This has meant that the reference numbers have changed throughout the manuscript.

Thank you again for reviewing our manuscript. Please let me know if you have any questions or require anything else.

Kind regards,

Kelly

---

## [Editor Report · Decision Letter 1]

20 Dec 2023

“Putting the power back into community”: A mixed methods evaluation of a chronic hepatitis B training course for the Aboriginal health workforce of Australia’s Northern Territory

PONE-D-23-20121R1

Dear Dr. Hosking,

We’re pleased to inform you that your manuscript has been judged scientifically suitable for publication and will be formally accepted for publication once it meets all outstanding technical requirements.

Kind regards,

Kimberly Page, PhD, MPH

Academic Editor

PLOS ONE
---

## [Editor Report · Acceptance letter]

12 Jan 2024

PONE-D-23-20121R1 

PLOS ONE

Dear Dr. Hosking, 

I'm pleased to inform you that your manuscript has been deemed suitable for publication in PLOS ONE. Congratulations! Your manuscript is now being handed over to our production team.

Kind regards, 

on behalf of

Dr. Kimberly Page 

Academic Editor

PLOS ONE